# A pair of non-Mendelian genes at the *Ga2* locus confer unilateral cross-incompatibility in maize

Zhibin Chen [1,2,3], Zhaogui Zhang [1,3], Huairen Zhang[1], Kai Li[1,2], Darun Cai [1,2], Li Zhao[1], Juan Liu [1✉] & Huabang Chen [1✉]

Maize unilateral cross-incompatibility (UCI) that causes non-Mendelian segregation ratios has been documented for more than a century. *Ga1*, *Ga2*, and *Tcb1* are three major UCI systems, described but not fully understood. Here, we report comprehensive genetic studies on the *Ga2* locus and map-based cloning of the tightly linked male determinant *ZmGa2P* and female determinant *ZmGa2F* that govern pollen-silk compatibility among different maize genotypes. Both determinants encode putative pectin methylesterases (PME). A significantly higher degree of methyl esterification is detected in the apical region of pollen tubes growing in incompatible silks. No direct interaction between ZmGa2P and ZmGa2F is detected in the yeast two-hybrid system implying a distinct mechanism from that of self-incompatibility (SI). We also demonstrate the feasibility of *Ga2* as a reproductive barrier in commercial breeding programs and stacking *Ga2* with *Ga1* could strengthen the UCI market potentials.

---

[1] State Key Laboratory of Plant Cell and Chromosome Engineering, Innovative Academy of Seed Design, Institute of Genetics and Developmental Biology, Chinese Academy of Sciences, Beijing 100101, China. [2] University of Chinese Academy of Sciences, 100039 Beijing, China. [3]These authors contributed equally: Zhibin Chen, Zhaogui Zhang. ✉email: juanliu@genetics.ac.cn; hbchen@genetics.ac.cn

Maize is a typical outcrossing and yet self-compatible species. Certain genotypes, however, hybridize in only one direction. This unilateral cross-incompatibility (UCI) has been known for more than a century and three major UCI systems, *Gametophyte factor1* (*Ga1*), *Gametophyte factor 2* (*Ga2*), and *Teosinte crossing barrier 1* (*Tcb1*), are well-documented[1–4]. *Tcb1* is predominantly present in teosinte, while *Ga1* and *Ga2* are in both maize and teosinte[3,5,6]. This one-way reproductive barrier is believed to play a role in restricting gene flow between maize and teosinte and among maize populations[7]. Similarities and uniqueness exist among *Ga1*, *Ga2*, and *Tcb1*. All three UCI loci are governed by a pair of female determinant that allows the silk to block non-self-type pollen fertilization and male determinant that overcomes the block of the female determinant from the same locus[8]. Accordingly, each UCI locus has three different haplotypes: *S*-haplotype possessing both male and female determinants, *M*-haplotype harboring only male determinant, and wildtype having neither[9]. During fertilization, the *S*-plants completely reject pollen from wildtype due to the female barrier but accept pollen of both *S*-plants and *M*-plants. The *M*-haplotype, which can reciprocally cross with the other two, serves as the cross-neutral mediator between *S*-haplotype and wildtype. The wildtype silks accept pollen from both *S*-haplotype and *M*-haplotype plants[5,10–12]. All three UCI systems serve as pre-zygotic barriers characterized by the retardation of pollen tube growth in incompatible silks[6]. Using a heteroallelic pollen approach, pollen-silk barriers to crossing in maize and teosinte were demonstrated to result from incongruity rather than active rejection[13]. *Ga1*, *Ga2*, and *Tcb1* are genetically distinct and cross-incompatible to each other. Although pollen tube growth is restricted in incompatible silks of *Ga1*, *Ga2*, and *Tcb1* plants, significantly different pollen tube morphologies were observed among the three UCI systems indicating different physiological and/or physiochemical mechanisms may exist among them[6].

To uncover the mystery of UCI, exploration of UCI genes is preliminarily required and major progress has been achieved in recent years[3,6–9,12,14,15]. A comparative de novo RNA-seq study of silks from near isogenic lines harboring *Ga1* and *ga1*, respectively, predicted *ZmPme3* as the candidate of the *Ga1* female determinant, which was the first report of PME involvement in maize UCI, but the identity has not yet been functionally validated[14]. Through a combination of map-based cloning, genome-wide association analysis, and Bacterial Artificial Chromosome (BAC) screening, Zhang reported the map-based cloning of the male determinant (*ZmGa1P*) of the *Ga1* locus. The identity of *ZmGa1P* was verified as the homozygous transgenic plants expressing *ZmGa1P* in a *ga1* background. These plants can fertilize *Ga1-S* plants and also can be fertilized by pollen of *ga1* plants. Further functional studies revealed that *ZmGa1P* encodes a pectin methylesterase (PME) predominantly localized to the apex of growing pollen tubes, which may interact with another pollen-specific PME protein ZmPME10-1 to maintain the state of pectin methylesterification required for pollen tube growth[8]. Immediately after this, the female determinant of *Tcb1* (*Tcb1-f*) also was map-based cloned with the aid of a de novo RNA-seq strategy as *Tcb1-f* was absent from the maize reference genome. Functional studies revealed that *Tcb1-f* also encodes a pectin methylesterase (PME38) and may prevent pollen tube growth by modifying cell wall of pollen tube[15]. All the findings demonstrated the critical role of PME activity in controlling maize UCI, probably by regulating the degree of methylesterification of the pollen tube cell wall[4].

Since the discovery of maize UCI phenomenon, research progress in this area has been slower than that in the self-incompatibility field, largely due to the difficulty of phenotyping and the genomic complexity of the maize UCI loci for mapping and map-based cloning[8]. No pair of the male and female determinants has been isolated for any maize UCI locus so far, which hindered the study of the specificity of each UCI locus and eventually deciphering of the molecular mechanism of maize UCI. Furthermore, the genetic nature of the female determinant has never been genetically investigated, and the initial designation of *Gametophyte factor* for maize UCI does not fully reflect the one-way reproductive barrier phenomena. Meanwhile, much less is known about the *Ga2* locus compared to the *Ga1* and *Tcb1* loci.

In this study, we report comprehensive genetic studies and map-based cloning of the male and the female determinants of the *Ga2* locus. We also demonstrate the feasibility of the utilization of *Ga2* as a reproductive barrier in maize breeding program and stacking *Ga1* and *Ga2* to strengthen the UCI application. Our work lays a foundation for future studies of maize UCI.

## Results

**Genetic studies of the *Ga2* locus.** To fully understand the genetic nature of the *Ga2* locus, we did a comprehensive genetic analysis of the male and the female determinants of the locus. The 511L (*Ga2*/*Ga2*) maize line was provided by the maizeGDB stock center (https://www.maizegdb.org/data_center/stock) and its UCI with B73, *Ga1*, and *Tcb1* was confirmed, respectively[16]. We also studied the *Ga2* locus distribution among a diverse set of 946 dent and flint maize lines[8] and found 2 *S*-haplotype (*Ga2-S/Ga2-S*) and 19 *M*-haplotype (*Ga2-M/Ga2-M*) inbred lines (Supplementary Data 1).

To study the male determinant genetically, (W22 × 511L) $F_1$ was developed using W22 (*ga2/ga2*) inbred line as female parent and 511L as male. 511L plants were then pollinated with the (W22 × 511L) $F_1$ pollen and full-seed set was observed, indicating the male determinant was of dominant nature. Next, 511L plants were pollinated by the individual plants of the W22 (♀) × (W22 × 511L) (♂) $BC_1F_1$ segregating population. At maturity, the ratio of plants setting full seeds and plants producing less than 5 seeds of the population was 1:1 by Chi-square independence tests (122:118, $\chi^2 = 0.033$, $P > 0.05$), demonstrating that the male determinant was conditioned by a single gene. To confirm the gametophytic nature of the male determinant, 511L (♀) × (W22×511L) (♂) $BC_1F_2$ segregating population were developed and 105 plants were randomly chosen to pollinate 511L plants. All 105 511L pollinated plants set full seeds, which proved that there were no *ga2/ga2* plants in the 511L (♀) × (W22×511L) (♂) $BC_1F_2$ segregating population and demonstrated the gametophytic nature of the male determinant. We also used two polymorphic DNA markers M1 and M9 between W22 and 511L flanking the *Ga2* locus to check the genotypes of 511L (♀) × (W22 × 511L) (♂) $BC_1F_1$ plants. A total of 88 individual plants were randomly selected for genotyping and only the alleles of the two markers from 511L were identified for all the plants tested (Supplementary Fig. 1a, b), further revealing that only pollen of the *S*-haplotype from 511L was able to pollinate 511L plants. The above genetic experiments confirmed the gametophytic nature of the male determinant of the *Ga2* locus.

To study the genetic nature of the female determinant of the *Ga2* locus, (W22 × 511L) $F_1$ plants were pollinated with W22 pollen and full-seed setting was obtained, indicating that the $F_1$ plants do not have the UCI function and its female determinant is of recessive nature. Meanwhile, individual plants of the (W22 × 511L) (♀) × 511L (♂) $BC_1F_1$ segregating population were pollinated with W22 pollen. The ratio of plants setting full seeds and plants setting less than 5 seeds was 1:1 by Chi-square independence tests (104:116, $\chi^2 = 0.328$, $P > 0.05$), proving that the female function also was conditioned by a single gene. To

determine whether the female determinant is gametophytic or sporophytic, a (W22 × 511L) (♀)×W22 (♂) BC$_1$F$_2$ segregating population was developed and 188 individual plants were randomly selected to be pollinated with W22 pollen. If the female was gametophytic, all the 188 plants would set full ears. In fact, the ratio of plants with full-seed setting and plants producing no seeds was 8:1 (167:21, $\chi^2 = 0.159$, $P > 0.05$), demonstrating the sporophytic nature of the female determinant. We also used two polymorphic markers M1 and M9 to genotype 88 (W22 × 511L) (♀) × W22 (♂) BC$_1$F$_1$ individuals randomly and found both $Ga2/ga2$ and $ga2/ga2$ genotypes at a 1:1 ratio, further confirming the sporophytic nature of the female determinant (Supplementary Fig. 2a, b). In conclusion, the $Ga2$ female determinant is of sporophytic nature and governed by a single recessive gene.

**Mapping and cloning of the male determinant**. In our previous study, we developed a homogeneous population mapping strategy to effectively map the male determinant at the $Ga1$ locus[9]. The same approach was applied to map the $Ga2$ male determinant. Briefly, 511l (♀) × (W22 × 511L) (♂) BC$_1$F$_1$ back-crossing population was developed. During the pollination process, pollen grains of $ga2$ were completely excluded from fertilization due to the gametophytic nature of the male determinant, and a homogeneous BC$_1$F$_1$ population was created in which only the $Ga2/Ga2$ genotype existed, making phenotyping unnecessary. A total of 16,544 seeds were screened for recombinants and the male determinant was narrowed to a 1.21-Mb region between markers M5 and M8 on chromosome 5 based on the B73_RefGen_v4[17] (Fig. 1a). Five genes were annotated (Fig. 1a) and only $Zm00001d016245$ was specifically expressed in 511L pollen (Supplementary Fig. 3a, b) and pollen grains of other $Ga2$-S and $Ga2$-M haplotypes (Supplementary Fig. 3c). Using $Zm00001d016245$ as genomic probe, the 511L BAC library was screened and a 136-kb BAC clone containing $Zm00001d016245$ was identified, sequenced, and assembled (Supplementary Data 2). In this BAC, two copies of $Zm00001d016245$ were identified, copy1 (G1) was intact while copy2 (G2) had a stop codon at the +229-position from the ATG start codon (Fig. 1b). $Zm00001d016245$ in B73 has a G deletion in the first exon leading to a premature stop codon at the +387-position compared to that of 511L (Fig. 1c and Supplementary Figs. 4, 5). $Zm00001d016245$ is then most likely the candidate gene of the male determinant.

To validate $Zm00001d016245$ has the male function, a 4993-bp genomic fragment was PCR-amplified from the 136-kb BAC clone containing a 2,664-bp promoter region upstream from the ATG start codon, a 1305-bp coding region, and a 1024-bp 3′ UTR region downstream from the TGA stop codon (Fig. 2a). The genomic fragment was then introduced into the maize line B104 ($ga2/ga2$) via $Agrobacterium$-mediated transformation and eight T$_0$ transgenic event lines were obtained. 511L plants pollinated with pollen of three event lines (MP1, MP2, and MP3) displayed full-seed settings at maturity, while 511L plants pollinated with pollen of the non-transgenic counterparts showed no seed setting (Fig. 2b). Moreover, the expression level of $Zm00001d016245$ in pollen of transgenic plants was consistent with the seed-setting results (Fig. 2c). Furthermore, the seeds were PCR-genotyped and confirmed as hybrids between 511L and the transgenic event lines (Fig. 2d). These results proved that $Zm00001d016245$ of 511L endowed $ga2$ pollen to have the male determinant function of the $Ga2$ locus. We designated $Zm00001d016245$ as $ZmGa2P$ for the $Ga2$ male determinant.

**Mapping and cloning of the female determinant**. To fine map the female determinant, (W22 × 511L) (♀)×511L (♂) BC$_1$F$_1$ population was developed in which $Ga2/Ga2$ and $Ga2/ga2$ were

segregating at a ratio of 1:1. The segregating individual plants were completely detasseled and pollinated with W22 pollen. A total of 2,600 individuals were genotyped and their seed-settings were recorded for recombinant screening. The female determinant was mapped in a 1.7-Mb region between markers M3 and M8 on chromosome 5 based on the B73_RefGen_v4 (Fig. 3a). There were eight annotated genes in the mapping region (Fig. 3a) and only $Zm00001d016248$ was specifically expressed in silks (Supplementary Fig. 6a). However, $Zm00001d016248$ expression level was low and no significant difference existed between 511L and B73 silks (Supplementary Fig. 6a). We then did a transcriptome comparison between 511L and B73 silks[16]. The transcripts were de novo assembled considering that the female determinant in $ga2$ may be a null allele. The most significantly upregulated (based on log$^2$ Fold Change) transcript in $Ga2$ silks was $TRINITY\_DN1207\_c0\_g1$ that was located within the 1.7-Mb mapping region (Supplementary Table 1 and Supplementary Data 3). Moreover, quantitative PCR (qPCR) analysis confirmed that $TRINITY\_DN1207\_c0\_g1$ was specifically expressed in the 511L silk tissues (Supplementary Fig. 6b). We then checked expressions of $TRINITY\_DN1207\_c0\_g1$ in the silks of 10 different maize inbred lines (two $Ga2$-S lines, four $Ga2$-M lines and four $ga2$ lines). The two $Ga2$-S lines showed high expression level while no expression was detected in both $Ga2$-M and $ga2$ lines (Supplementary Fig. 6c). Using $TRINITY\_DN1207\_c0\_g1$ as probe, a 133-kb BAC clone was identified, sequenced and assembled from the 511L BAC library (Supplementary Data 4). The BAC was well-aligned with B73 sequence in the mapped region and contained two annotated sequences, one was $Zm00001d016242$ ($PG1$) that was not expressed in silks (Supplementary Fig. 6a) and the other was the transcript $TRINITY\_DN1207\_c0\_g1$ ($PG2$) (Fig. 3b). Major differences between 511L and B73 existed in the sequences of $TRINITY\_DN1207\_c0\_g1$. B73 has an A insertion in the first exon at the +94-position from the ATG start codon, which causes a premature termination (Supplementary Figs. 7, 8) and a 33,858-bp LTR (long-terminal-repeat retrotransposon) insertion at the −423-position in the promoter region (Fig. 3c). There also was a 11,757-bp deletion including 265-bp of exon 2 and 11,492-bp 3′ UTR region (Fig. 3c). Transcript $TRINITY\_DN1207\_c0\_g1$ is most likely the candidate of the female determinant.

To validate $TRINITY\_DN1207\_c0\_g1$ has the female function, a genomic DNA fragment containing 2052-bp promoter, 1288-bp coding sequence and 1092-bp terminator region from 511L was introduced into the maize inbred line B104 ($ga2/ga2$) (Fig. 4a). Eight T$_0$ events were developed and selfed to generate T$_1$ lines. Non-transgenic T$_1$ lines were eliminated by herbicide application at seedling stage. If 511L $TRINITY\_DN1207\_c0\_g1$ had the female determinant function, homozygous transgenic T$_1$ lines were expected not able to produce seeds by selfing since B104 ($ga2/ga2$) does not have the male determinant. To overcome this challenge, a purple kernel line ZYM1 ($ga2/ga2$) and a yellow kernel $M$-haplotype line Mo17[18] ($Ga2$-M/$Ga2$-M lacking the female determinant) were employed. At silking stage, the T$_1$ transgenic-positive plants were first pollinated with ZYM1 pollen and then pollen of Mo17. If $TRINITY\_DN1207\_c0\_g1$ was the female determinant, homozygous transgenic plants would produce yellow kernels (excluding ZYM1 pollen but accepting Mo17 pollen) while heterozygous plants set both purple and yellow kernels (accepting both pollen). Three T$_1$ transgenic lines showed both complete yellow kernel ears and purple-yellow mixed-colored ears at maturity (Fig. 4b). Transgene expression analysis showed significantly higher levels in silks of the plants producing yellow kernels (Fig. 4c). These results indicated that plants with mixed-colored kernels were lacking UCI function and plants with complete yellow kernels were able to exclude $ga2$

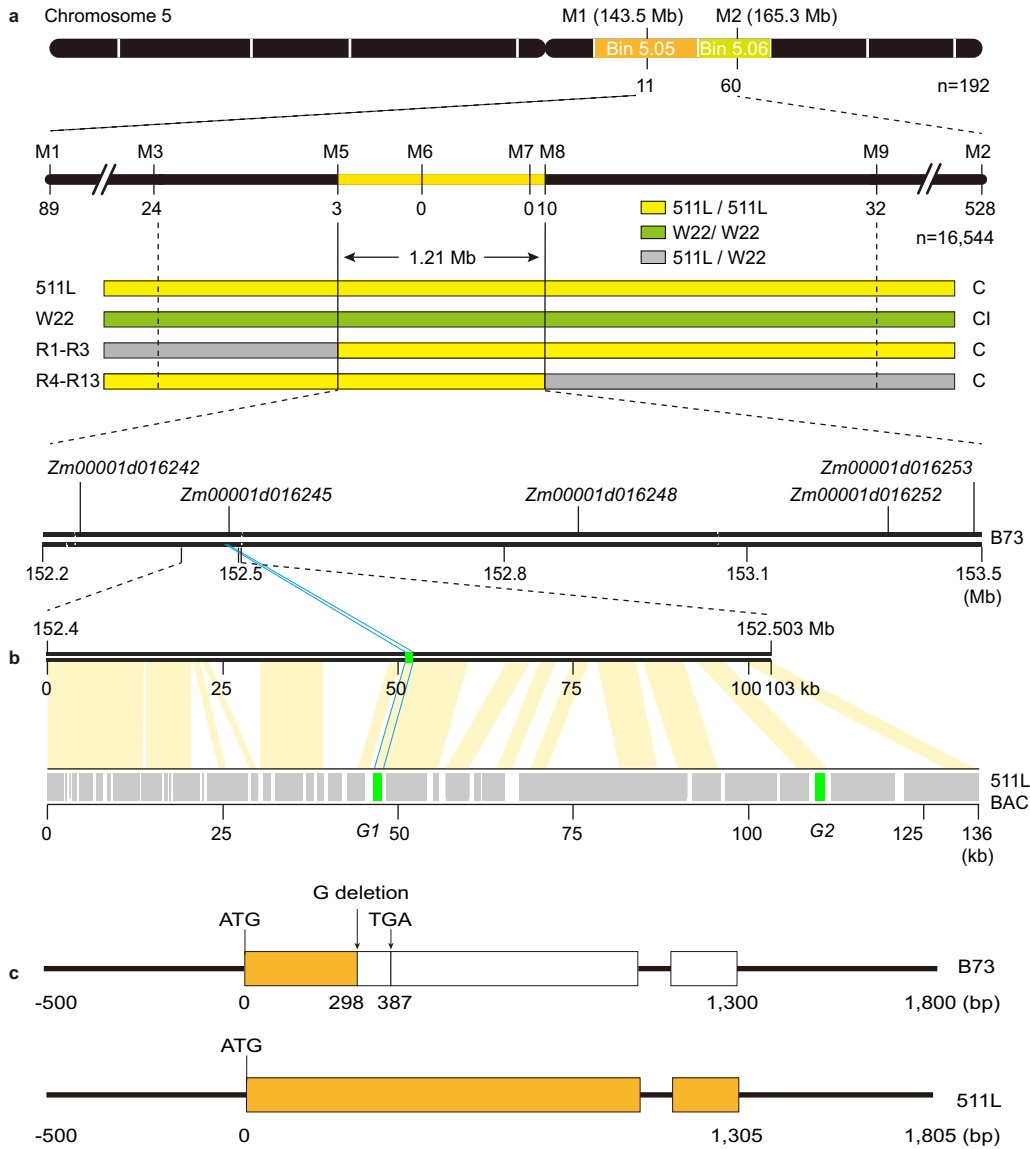

**Fig. 1 Map-based cloning of the male determinant. a** The male determinant was fine-mapped to a region between markers M5 and M8 containing five annotated genes. Recombinants were indicated below the markers. R1-R3, recombinants. C, cross-compatible; CI, cross-incompatible. **b** The BAC clone harboring *Zm00001d016245* was aligned to the male determinant mapping region. Two potential genes (*G1* and *G2*; green color) were predicted in the BAC sequence using the FGENESH gene annotation program. The gray boxes indicated the locations of transposons and retrotransposons. The orange lines represented synteny regions between BAC sequence and its homologs in B73. **c** Gene structure of *Zm00001d016245* in B73 and 511L.

pollen but accept pollen of the *M*-haplotype. The $T_1F_1$ yellow kernels were 100% herbicide resistant demonstrating the transgenic female parental plant was homozygous at the transgene locus. The $T_1F_1$ plants of the yellow kernels were selfed (due to pollen of the *M*-haplotype of Mo17) and $T_1F_2$ were generated. Non-transgenic plants of the $T_1F_2$ population were eradicated by herbicide spray and transgenic ones were both selfed and pollinated with pollen of ZYM1. Yellow-kernel ears, purple-yellow mixed-colored ears and ears with no seed-setting (Fig. 4d) showed up as expected. Collectively, these results demonstrated that homozygous transgenic plants of the transgene *TRINITY_DN1207_c0_g1* were able to exclude *ga2* pollen but accept pollen of *Ga2*. We designated *TRINITY_DN1207_c0_g1* as *ZmGa2F* for the female determinant of the *Ga2* locus.

**ZmGa2P and ZmGa2F are putative PMEs mediating cross-incompatibility.** Sequence alignments of *ZmGa2P* and *ZmGa2F* revealed that both were predicted to encode putative PMEs that

have been reported to play critical roles in regulating pollen tube growth[19,20]. Abnormal pollen tube growth was observed in the incompatible silks of *Ga2* plants when pollinated with *ga2* pollen[6,16]. To better study the *Ga2* locus, near isogenic lines Zheng58$^{Ga2-S}$ and Zheng58$^{ga2}$ were developed by marker-assisted backcrossing for eight generations. Three compatible crosses Zheng58$^{Ga2-S}$/Zheng58$^{Ga2-S}$, Zheng58$^{ga2}$/Zheng58$^{Ga2-S}$, and Zheng58$^{ga2}$/Zheng58$^{ga2}$, and one incompatible cross Zheng58$^{Ga2-S}$/Zheng58$^{ga2}$, were made, respectively. PMEs catalyze pectin deesterification and the antibody LM20 preferentially recognizes methylesterified pectin[21]. To visualize the in vivo difference in the degree of methyl esterification (DM) at the pollen-tube-silk interface, the pollinated silk sections of the four different crosses were immunolabeled with LM20. There was no significant difference in the fluorescent signal intensity among the apical regions of the pollen tubes growing in the three compatible silks (Fig. 5a, b). Significantly elevated fluorescent signals were detected in the apical region of *ga2* pollen tubes growing in the

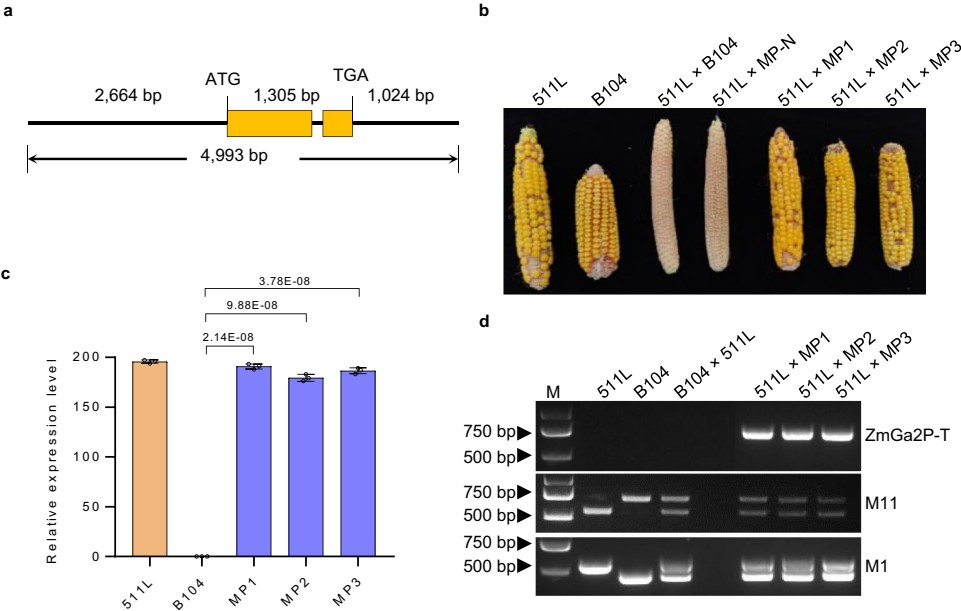

**Fig. 2 Transgenic validation of the male determinant. a** Structure of *Zm00001d016245* used for the complementation assay. **b** Cross-compatibility analysis. Ears represented selfed, reciprocal crosses between *Ga2-S* (511L) and *ga2* (B104), and 511L pollinated with non-transgenic controls (MP-N) and transgenic plants (MP1, MP2, and MP3) expressing *Zm00001d016245* gene. Each cross contained at least 3 ears. **c** Expression analysis of *Ga2-S* type *Zm00001d016245* in pollen of the indicated plants. Expression means and Standard Errors were from 3 biological replicates (unpaired two-tailed Student's *t*-test). The *P* values are indicated in the graphs. **d** Genotyping of the cross-pollinated seeds. Marker Ga2P-T was used to confirm the transgenes; markers M1 and M11 were used to check the heterozygosity. M is DNA marker. Source data are provided as a Source data file.

incompatible Zheng58$^{Ga2-S}$ silks (Fig. 5a, b), suggesting high DM pectin presenting in the *ga2* pollen tubes upon growing in *Ga2-S* silks. High DM may result from compromised PME activity and/or the impact of the female determinant on modifying methylesterified pectin levels of pollen tube wall.

In the well-studied SI systems, the male and the female determinants/proteins of the S-LOCUS directly interact to regulate SI[22], which prompted us to hypothesize that the two determinants at *Ga2* locus may interact. We then performed yeast two-hybrid experiments to investigate the interaction, but no direct interaction between ZmGa2F and ZmGa2P was detected (Supplementary Fig. 9). Since the degree of methyl esterification of pollen tube was significantly altered in pollen tubes incompatible crosses, other components including PMEs in pollen and/or silk of UCI plants were most likely to be involved. In our previous study, ZmPME10-1, a type-I/group 2 pollenspecific PME having both PME and PMEI domains, was identified to be specifically expressed in large amount in pollen of all maize genotypes[8]. We found that ZmGa2P interacted with both the PME domain and the full length of ZmPME10-1, while ZmGa2F only interacted with the PME domain of ZmPME10-1 in yeast (Supplementary Fig. 9), implying that ZmGa2P may have competitive advantage over ZmGa2F when interacting with ZmPME10-1.

**Application of the *Ga2* locus in maize breeding programs**. We previously reported the feasibility of the utilization of the *Ga1* locus in isolating undesired pollen in commercial maize production[8]. The fact that the majority of dent and flint maize varieties carry the *ga2* haplotype also makes *Ga2* a potential reproductive barrier. To validate the commercial value of the *Ga2* locus in this field, the *Ga2* locus was introduced into two *ga2* lines of Zheng58$^{ga2}$ and Chang7-2$^{ga2}$, respectively. The hybrid (Zheng58$^{ga2}$/Chang7-2$^{ga2}$) has been the most planted variety for the past 15 years in China. Based on field tests in Beijing and Sanya, Hainan province, hybrids of Zheng58$^{ga2}$/Chang7-2$^{ga2}$ and

Zheng58$^{Ga2-S}$/Chang7-2$^{Ga2-S}$ showed no significant difference in agronomic traits, but Zheng58$^{Ga2-S}$/Chang7-2$^{Ga2-S}$ completely excluded *ga2* pollen from fertilization (Fig. 6a and Supplementary Table 2), proving *Ga2* also a feasible commercial reproductive barrier.

Although the majority of dent and flint maize lines carry *ga1* and *ga2* haplotypes, S-haplotypes and M-haplotypes, especially M-haplotypes, were identified among elite maize lines[8,18] (Supplementary Data 1), which greatly compromised the utilization of *Ga1* and *Ga2* as effective reproductive barriers in commercial maize breeding programs. We then stacked *Ga1* with *Ga2* and developed homozygous lines at both loci, and four such lines were bred. The four lines behaved the same at two different locations for two years, in which they only accepted pollen harboring both *Ga1* and *Ga2* loci, but were completely crossincompatible to pollen of *Ga1*, *Ga2*, and pollen mixture of *Ga1* and *Ga2*, respectively (Fig. 6b). This pyramiding strategy can overcome the risks imposed by the S-haplotype and M-haplotype maize lines and could have significant market potentials.

## Discussion

Maize UCI was first reported in the early 1920s and the term *gametophyte factor* (*Ga*) was employed to describe the locus[1,23]. This term, however, does not fully describe these loci since more and more studies have come to a consensus that the three major UCI systems, *Ga1*, *Ga2*, and *Tcb1*, are regulated by a male-female gene pair that are genetically tightly linked. The male determinant of *Ga1* (*ZmGa1P*) and the female determinant of *Tcb1* (*Tcb1-f*) have been functionally validated[8,15]. Homozygous *ZmGa1P* transgenic plants did not exhibit the female determinant function. The fact that the M-haplotype lacking the female determinant exists in natural maize populations suggests the break-down of the male-female determinant pair via recombination although the frequency is low. Another possibility is that the female determinant mutated leading to the M-haplotype. No line with only female determinant has ever been identified since such a line

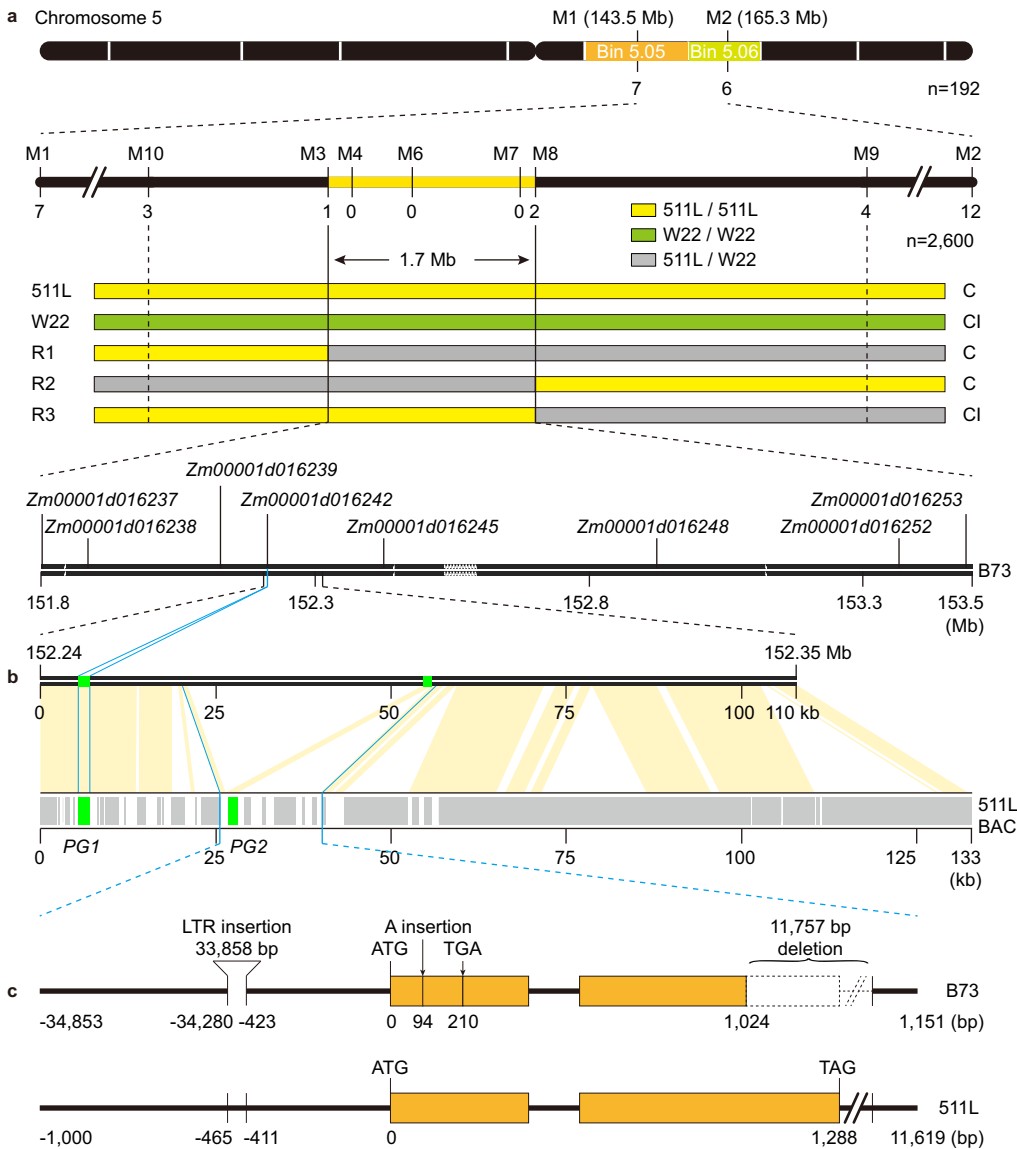

**Fig. 3 Map-based cloning of the female determinant. a** The female determinant was narrowed to a region between markers M3 and M8 containing eight annotated genes. Recombinants were indicated below the markers. R1-R3, recombinants. C, cross-compatible; CI, cross-incompatible. **b** The BAC clone harboring *TRINITY_DN1207_c0_ g1* was aligned to the female determinant mapping region. Two potential genes (*PG1* and *PG2*; green color) were predicted in the BAC sequence using the FGENESH gene annotation program. The gray boxes indicated the locations of transposons and retrotransposons. The orange lines represented synteny regions between BAC sequence and its homologs in B73. **c** Gene structure of *TRINITY_DN1207_c0_ g1* (*PG2*) in B73 and 511L.

would lack the male determinant and be self-incompatible. The male and the female determinants of the *Ga2* locus were mapped in the same region of 1.7-Mb on chromosome 5, indicating their tight linkage. Nevertheless, no male-female determinant pair at any locus has been cloned so far largely due to the genomic complexity of the UCI loci. In the cases of *Ga1* and *Ga2*, the mapping region could not be narrowed to a single gene, even when a large number of individuals were screened for recombinants. Preliminary studies on *Ga1, Ga2,* and *Tcb1* revealed a PME-mediated mechanism to control maize UCI. Tcb1-f/PME38, the female determinant of *Tcb1*, and ZmPme3, a candidate for *Ga1* female determinant, were shown to differ in eight amino acids and diverged ~175,000 years ago, well before the split between the *mexicana* and *parviglumis* subspecies of teosinte[4]. We found that ZmGa2F shared 47.29% and 47.92% in amino acid identity with ZmPme3 and Tcb1-f/PME38 (Supplementary Fig. 10a), respectively, which may suggest that the *Ga2* locus

diverged well before the split of the *Ga1* and *Tcb1* loci. The *Ga1* locus was likely a duplicate of the *Tcb1* locus based on the fact that they both were located on the same chromosome 4 with a genetic distance of 44 cM[3]. The cloning of the male and female determinants at the *Ga2* locus opens the door for studying direct interactions of the male-female determinant pair and molecular mechanisms in UCI specificity among different loci.

Self-incompatibility (SI) has been well-studied in many plant taxa including Solanaceae, Rosaceae, Plantaginaceae, Brassicaceae, and Papaveraceae[24–27]. Although maize UCI and SI share common features in that they both are pre-zygotic reproductive barriers and governed by tightly linked male and female determinants, there exist significant differences that distinguish maize UCI from SI. In gametophytic SI in Papaveraceae, the female determinant PrsS interacts with the male determinant PrpS leading to programmed cell death[28–30]. The female SI protein S-LOCUS RECEPTOR KINASE (SRK) and the male protein

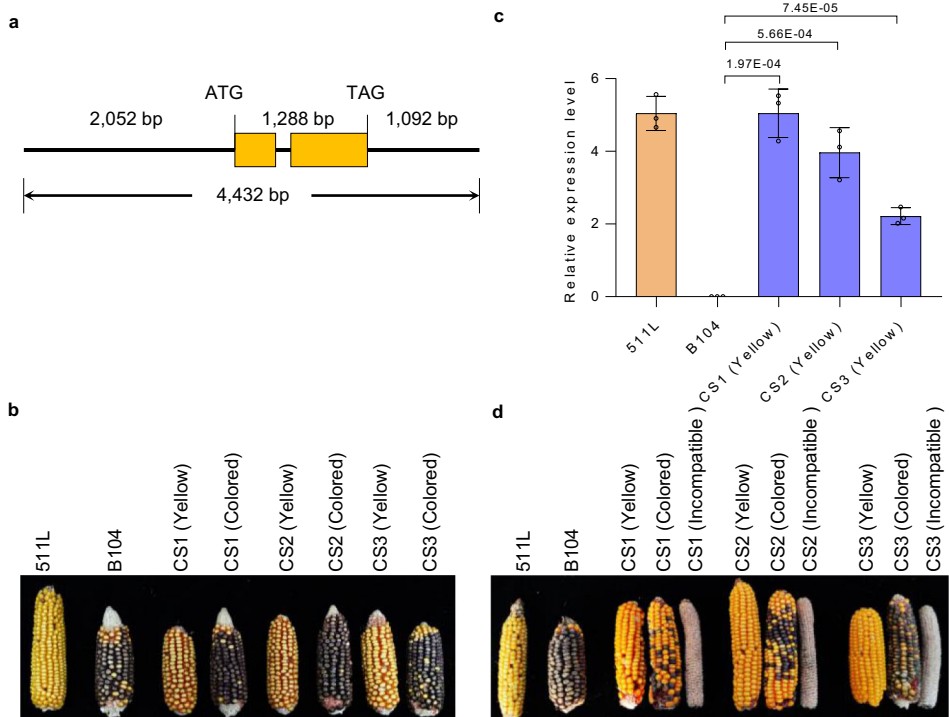

**Fig. 4 Transgenic validation of the female determinant. a** Structure of *TRINITY_DN1207_c0_ g1* used for the complementation assay. **b** Cross-compatibility analysis. Ears of $T_1$ transgenic plants pollinated with ZYM1 (*ga2/ga2*, purple kernel) and Mo17 (*Ga2-M/Ga2-M*, yellow kernel). CS1, CS2, and CS3 represent transgenic plants that expressed *TRINITY_DN1207_c0_ g1* in B104. Each cross contained at least 3 ears. **c** Expression analysis of *Ga2-S* type *TRINITY_DN1207_c0_ g1* transcript in silks of the indicated plants. Expression means and Standard Errors were from 3 biological replicates (unpaired two-tailed Student's *t*-test). The *P* values are indicated in the graphs. **d** Cross-compatibility analysis. Ears of $T_1F_2$ transgenic plants pollinated with ZYM1 and selfed plants. Each cross contained at least 3 ears. Source data are provided as a Source data file.

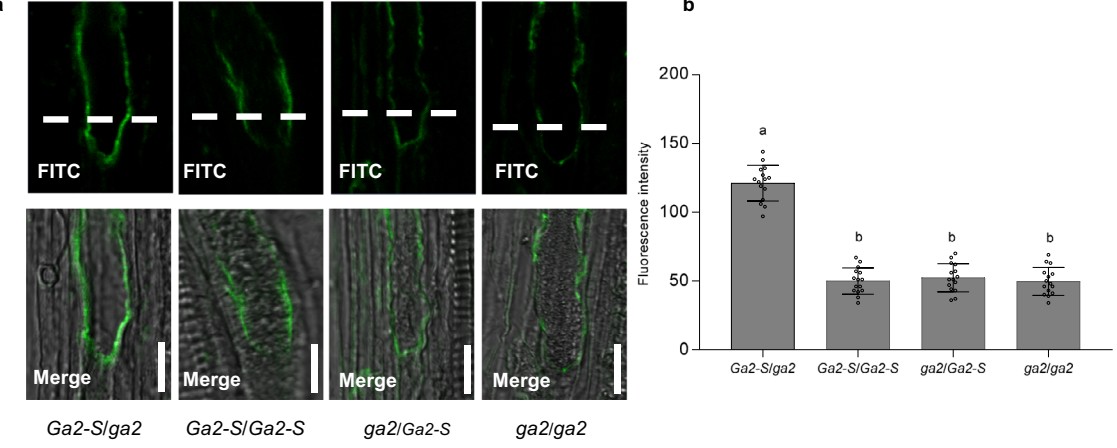

**Fig. 5 Immuno-detecting of the degree of methylesterification at the apical region of pollen tubes. a** Pectin methylesterification of longitudinal sections of pollen tubes in near isogenic lines was immunodetected with the LM20 antibody. The fluorescent signal intensity was detected in Zheng58$^{Ga2-S}$/ Zheng58$^{ga2}$ (*Ga2-S/ga2*), Zheng58$^{Ga2-S}$/Zheng58$^{Ga2-S}$ (*Ga2-S/Ga2-S*), Zheng58$^{ga2}$/Zheng58$^{Ga2-S}$ (*ga2/Ga2-S*) and Zheng58$^{ga2}$/Zheng58$^{ga2}$ (*ga2/ga2*). The fluorescent signal intensity of *ga2* pollen tubes was significantly elevated in Zheng58$^{Ga2-S}$ pollen, while the fluorescent signal in three compatible crosses showed no significant difference. The dotted lines are the arbitrary division lines between the apical region and the rest of the pollen tube. The immunodetected fluorescence signal intensity of apical region of the pollen tube (below dotted lines) were quantified. FITC is fluorescein isothiocyanate. Scale bars = 10 μm. **b** Quantitative analysis of immunodetected fluorescence signal intensity in (**a**). Data means and Standard Errors were derived from statistical values from 15 pollen tubes. a and b indicate the significant difference of data means(one-way ANOVA Tukey's multiple comparisons test). Source data are provided as a Source data file.

S-LOCUS PROTEIN11/S-LOCUS CYSTEINE-RICH (SP11/SCR) interact to initiate a signal transduction cascade leading to self-pollen growth arrest in sporophytic SI in Brassicaceae[31–37]. In Solanaceae, the female S-RNase interacts with the male S-LOCUS F-box protein to govern SI[34,38–42]. Therefore, direct interaction

between male and female determinants was the unique feature of SI system. Several lines of evidence, however, imply that direct interaction may not exist between the male and the female determinants of the maize UCI. First, the male determinants of *Ga1* and *Ga2*, the female determinants of *Ga2* and *Tcb1*, and the

**Fig. 6 Application of the *Ga2* locus in maize breeding program and stacking of *Ga1* and *Ga2*. a** Zheng58$^{Ga2-S}$/Chang7-2$^{Ga2-S}$ (*Ga2-S/Ga2-S*) and its near isogenic hybrid Zheng58$^{ga2}$/Chang7-2$^{ga2}$ (*ga2/ga2*) were pollinated with pollen mixture of a *ga2* line ZYM1 (purple kernel) and the hybrid of Zheng58$^{Ga2-S}$/Chang7-2$^{Ga2-S}$ in Beijing, China. **b** Compatibility test of double homozygous maize lines at *Ga1* and *Ga2* loci. *Ga1Ga2* double homozygous plants were selfed (*Ga1/Ga2*) and pollinated with pollen of SDGa25 (*Ga1*), 511L (*Ga2*), pollen mixture of SDGa25 and 511L (Ga1 + Ga2), and Zheng58 (*ga2*), respectively. Each cross contained at least 3 ears.

putative female determinant of *Ga1*, all encoded the type-II/group 1 PME proteins. This type of protein does not have the PMEI domain that was proved to interact with PME to regulate PME activities[19,43–47]. Second, no direct interaction between ZmGa2P and ZmGa2F was detected in yeast two-hybrid system. Third, crystal structure of plant PME revealed five residues, Q113, Q135, D136, D157, and R225, were highly conserved, and mutations at these positions significantly reduced PME activities[19,44]. Various mutations of the five conserved residues were identified in maize UCI-related PMEs (Supplementary Fig. 11), suggesting they may not possess in vivo PME activities, and yet may directly interact with other active PMEs that have both PME and PMEI domains to finely tune PME activities in the apical region of pollen tube. The male and the female determinants at *Ga1*, *Ga2* and *Tcb1* loci were predicted to contain a signal peptide (Supplementary Fig. 12), which hinted that they may be secret extracellularly and modulate the final state of pectin methylation in pollen tube cell walls[4,48]. Furthermore, we observed similar fluorescent signal intensity among the apical regions of the pollen tubes of Zheng58$^{Ga2-S}$ and Zheng58$^{ga2}$ growing in Zheng58$^{ga2}$ silks (both were cross-compatible), implying ZmGa2P may not have PME activity. Fluorescent signals in the apical region of *ga2* pollen tubes growing in incompatible Zheng58$^{Ga2-S}$ silks were significantly elevated compared to those of *Ga2* pollen tubes, suggesting that other PMEs that interacted with ZmGa2P and/or ZmGa2F were involved in the control of pectin configuration. Similar results were obtained at the *Ga1* locus[8]. ZmPME10-1 was previously identified as an abundant pollen-specific type-I/group 2 PME and expected to play key roles in pollen tube growth[8]. Dysfunction of VANGUARD1, a homolog of ZmPME10-1 isoform in *Arabidopsis*, caused abnormal pollen tube growth and male sterility[20]. ZmGa2P was found to interact with both the full length and the PME domain of ZmPME10-1. Interestingly, ZmGa2F only interacted with the PME domain of ZmPME10-1. We hypothesized that ZmGa2P had competitive advantages over ZmGa2F to interact with ZmPME10-1. In an incompatible cross with *Ga2* plants as females, normal function of ZmPME10-1 was blocked by ZmGa2F resulting in pollen tube growth arrest. In a compatible cross, ZmGa2P-ZmPME10-1 complex was formed, ZmPME10-1 function was protected, and the impact of ZmGa2F on ZmPME10-1 was substantially minimized. Interestingly, ZmGa1P also interacted with ZmPME10-1[8] although ZmGa1P and ZmGa2P shared only 54.52% amino acid identity (Supplementary Fig. 10b). Whether ZmPME10-1 is involved in maize UCI regulation and how UCI specificity is determined remain unclear. ZmPME10-1 knockout lines are being developed to help address the intrinsic mechanism of PME-mediated UCI in maize.

Both *Ga1* and *Ga2* have been demonstrated as effective reproductive barriers in isolation among maize varieties of different kinds in commercial breeding programs. However, this practice is being challenged by the presence of the *M*-haplotypes at *Ga1*, *Ga2*, and *Tcb1* loci. Stacking *Ga1* with *Ga2* proved efficient to prevent pollination with the *M*-haplotypes. Pyramiding of *Ga1*, *Ga2* and *Tcb1* and development of lines homozygous at three loci are under way.

## Methods

**Plant materials and growth conditions**. The maize inbred line 511L, harboring *Ga2-S*, was obtain from MaizeGDB stock center (https://www.maizegdb.org/data_center/stock?id=9020818). All the maize lines used in this study, including the mapping populations, were planted in Beijing (39.5°N, 116.4°E) and San Ya (18.3°N, 108.8°E), Hainan province. The transgenic plants were grown in greenhouse with a 16-h light (25 °C)/8-h dark (18 °C) cycle[8]. All plants were planted in plots at 0.6 m (horizontal) × 0.4 m (longitudinal).

**Genetic analysis of the male determinant of the *Ga2* locus**. To study the genetic nature of the male determinant, (W22 × 511L) $F_1$ was developed using W22 as female parent and 511L as male. Full-seed set was observed in $F_1$, indicating the male determinant was of dominant nature. W22 was then pollinated with the (W22 × 511L) $F_1$ to generate the W22 (♀) × (W22 × 511L) (♂) $BC_1F_1$ segregating population, and the detasseled 511L plants were pollinated by the $BC_1F_1$ individuals to test the phenotypes. At maturity, the ratio of plants setting full seeds and plants producing less than 5 seeds were counted and statistically analyzed to confirm the single gene nature of the male determinant. To elucidate the gametophytic nature of the male determinant, 511L (♀) × (W22 × 511L) (♂) $BC_1F_1$ population was developed using 511L as female parent and $F_1$ (W22 × 511L) as male. If the male determinant was gametophytic, only one genotype (*Ga2/Ga2*) would be present in the 511L (♀) × (W22 × 511L) (♂) $BC_1F_1$ population, otherwise if it were sporophytic then two genotypes (*Ga2/Ga2* and *Ga2/ga2*) would be present. Next, we generated a 511L (♀) × (W22 × 511L) (♂) $BC_1F_2$ population by selfing the 511L (♀)×(W22 × 511L) (♂) $BC_1F_1$ plants and 105 plants were randomly chosen to pollinate the detasseled 511L plants. All 105 511L pollinated plants set full seeds, meaning that there were no *ga2/ga2* plants in the 511L (♀) × (W22 × 511L) (♂) $BC_1F_2$ population. Furthermore, we checked the genotypes of 88 plants randomly selected from the 511L (♀) × (W22 × 511L) (♂) $BC_1F_1$ population using polymorphic markers M1 and M9 flanking the *Ga2* locus. Only the alleles of 511L were detected, further confirming the gametophytic nature of the male determinant.

**Genetic analysis of the female determinant of *Ga2* locus**. We pollinated W22 to detasseled (W22 × 511L) $F_1$ to investigate the genetic nature of the female determinant. All 5 $F_1$ plants were full seed-set meaning that the female determinant was of recessive nature since $F_1$ lost the pistil barrier. A (W22 × 511L) (♀) × 511L (♂) $BC_1F_1$ segregating population was generated and pollinated with W22 to check the phenotypes. The ratio of plants setting full seeds and plants producing less than 5 seeds were counted and statistically analyzed to confirm the single-gene nature of the female determinant. To figure out whether the female determinant is gametophytic or sporophytic, we developed a (W22 × 511L) (♀) × W22 (♂) $BC_1F_1$ population. If the female determinant was gametophytic in nature, the $BC_1F_1$ population should have only one genotype *ga2/ga2*, otherwise if were sporophytic, there would be two genotypes *Ga2/ga2* and *ga2/ga2*. We then selfed the (W22 × 511L) (♀)×W22 (♂) $BC_1F_1$ plants to obtain a $BC_1F_2$ population, from which 188 individuals were randomly selected, detasseled and pollinated with W22. By Chi-square independence tests, an 8:1 ratio of plants setting full seeds and plants producing less than 5 seeds was observed, revealing the sporophytic nature of the female determinant. Moreover, a total of 88 randomly selected individuals

from (W22 × 511L) (♀)×W22 (♂) BC₁F₁ population were genotyped using M1 and M9. The presence of both $Ga2/ga2$ and $ga2/ga2$ confirmed the sporophytic nature of the female determinant.

**Fine mapping of the male determinant of Ga2 locus**. Due to the gametophytic nature of the male determinant, a homogeneous mapping population 511L (♀) × (W22 × 511L) (♂) was developed using 511L as female and (W22 × 511L) F₁ as male. A total of 16,544 individuals were screened for recombinants with the molecular markers listed in Supplementary Table 3. The male determinant was finally narrowed to a 1.21-Mb region (referred to B73_RefGen_V4 sequence) between M5 and M8.

**Fine mapping of the female determinant of Ga2 locus**. We first generated a (W22 × 511L) (♀) × 511L (♂) BC₁F₁ segregating population by pollinating (W22 × 511L) F₁ with 511L. The detasseled BC₁F₁ individuals were then pollinated with W22 pollen for phenotyping. A total of 2600 plants were screened for recombinants using the molecular markers listed in Supplementary Table 3. The female determinant was finally narrowed to a 1.7-Mb region (referred to B73_RefGen_V4 sequence) between M3 and M8.

**Construction, screening, and sequencing of the 511L BAC library**. To construct a BAC library of 511L, high quality genomic DNA was extracted from etiolated seedlings and was digested by $Hind$ III and separated by 1% pulsed field gel electrophoresis. The digested DNA fragments were retrieved and ligated with pIndigoBAC536-S vector before transformed into $E. coli$ DH10B-T[49,50]. By screening on LB medium with chloramphenicol (12.5 mg/L), 5-bromo-4-chloro-3-indolyl-β-D-galactopyranoside (80 mg/L) and isopropyl β-D-1-thiogalactoside (100 mg/L), white clones were individually picked into 384-well plates. There are totally 207,360 BAC clones arranged in 540 384-well plates, representing a 10-fold coverage of the maize genome. We developed a PCR-based method to screen clones harboring the male and female determinants, respectively and extracted plasmid DNA using PhasePrep™ BAC DNA Kit (Sigma-Aldrich, St. Louis, MO, USA). BAC DNA sequencing was performed using PacBio SMRT sequencing technology by the Nextomics Biosciences Company (Wuhan, China)[51] and the reads were de novo assembled with HGAP v2.3.0[52]. Eventually, a 136,543-bp contig harboring the male determinant and a 132,840-bp contig harboring the female determinant were obtained. Annotation was performed using the FGENESH gene annotation program (www.softberry.com)[53]. Interspersed repeats and low complexity DNA sequences were screened using the RepeatMasker program (http://www.repeatmasker.org/). Primers are listed in Supplementary Table 4.

**Gene expression analysis**. The samples of pollen, silks, roots, stems and leaves were collected at the proper growth stages for RNA isolation. Total RNA from the different maize tissues was isolated using TRIzol (Invitrogen) and treated with RNase-free DNase I (Thermo Fisher Scientific) to remove DNA according to the manufacturer. Complementary DNA (cDNA) was synthesized from 2 μg of total RNA using RevertAid reverse transcriptase (Thermo Fisher Scientific) with the oligoT₁₈ primer. To examine the expression level of the candidate genes, qPCR was performed on a LightCycler 480 Real-Time PCR System (Roche Diagnostics) with SYBR Green Master mix (45 cycles of 95 °C for 10 s, 60 °C for 30 s, and 72 °C for 10 s). Three biological and technical replicates were tested. $ZmGAPDH$ was used as an internal control to calculate the relative expression level using the $2^{-\Delta Ct}$ method. Primers are listed in Supplementary Table 4.

**Transcriptome de novo assembly and DEGs identification**. Silks of 511 L and B73 were collected for RNA-Seq analysis, the procedures of sampling, library construction and sequencing were based on the study[16]. Briefly, the unpollinated silks were collected at flowing stage and extracted to perform RNA-seq. Subsequently, sequencing works were performed by the Berry Genomics Company (Beijing, China) on an Illumina HiSeq 2500 V4 platform. The quality control of raw data was processed by Fastp software[54], which cut the first 15-bp of each read from 5' to 3' and removed low-quality reads and adapters. Trinity software[55] was used to de novo assemble the transcriptome, and the default parameters were used in the command line. The transcripts abundances were computed by Salmon software[56]. The DEGs between samples were identified by DESeq2 package[57] in R. Differential expressed gene (DEGs) were classified as having an adjusted $p$-value ≤ 0.01 and |log² Fold Change | ≥ 2 after correction for multiple testing.

**Generation of maize transformation plants**. To confirm $Zm00001d016245$ as the $Ga2$ male determinant, a 4,993-bp genomic fragment containing the coding region, the 2664-bp promoter region and 1024-bp terminator regions from 511L was inserted into the vector pCAMBIA3300[8], to generate the transformation construct. The resulting construct was transformed into the wildtype inbred line B104 through $Agrobacterium tumefaciens$ infection. We obtained total eight positive transgenic events and selfed to generate T₁ individuals, which then pollinated at least five 511L plants to verify the cross-incompatibility. Positive transgenic events were identified using the primers ZmGa2P-T and M1 and M11 were used to detect the pollution of undesired pollen.

For the genomic complementary assay of $TRINITY\_DN1207\_c0\_g1$, a genomic DNA fragment containing 2052-bp promoter, 1288-bp coding sequence and 1092-bp terminator region from 511L was introduced into the wildtype inbred line B104. Eight transgenic positive lines were obtained and selfed to generate T₁ individuals. Positive transgenic plants were screened with 5% basta (10% Glufosinate, Coolaber, Beijing) at the seedling stage. Supposing that $TRINITY\_DN1207\_c0\_g1$ had the female function, the homozygous transgenic T₁ lines were self-incompatible since B104 had no male function to overcome the pistil barrier. Then a purple kernel line ZYM1 ($ga2/ga2$) and a yellow kernel $M$-haplotype line Mo17 ($Ga2$-$M/Ga2$-$M$) were used to pollinate the transgenic T₁ plants. At flowering stage, T₁ individuals were first pollinated with ZYM1 and then Mo17. If $TRINITY\_DN1207\_c0\_g1$ was the female determinant, homozygous transgenic plants should produce yellow kernels while heterozygous plants set both purple and yellow kernels. As expected, three T₁ transgenic-positive lines produced either complete yellow kernels or a mix of colored seeds. Moreover, the T₁F₁ yellow kernels showed 100% basta-resistance, indicating that the transgenic female parental plant was homozygous. To obtain the self-compatible transgenic lines, we planted the T₁F₁ yellow kernels and selfed to generate T₁F₂. In T₁F₂ populitaions, the non-transgenic plants were eliminated by herbicide screening and the transgenic ones were selfed and pollinated with pollen of ZYM1 at maturity. Owing to the huge divergence of flowering time among the T₁F₂ individuals, two pollination procedures were adopted, one is selfed after pollinated with ZYM1 and the other is a single pollination with a mixture pollen of both pollen genotypes. Besides the full yellowed and the mixed color ears, barren ears were also observed due to some individuals which lack the male determinant.

**Generation of near isogenic lines at the Ga2 locus**. To generate near isogenic lines, Zheng58$^{ga2}$ and Chang7-2$^{ga2}$ were first pollinated with 511L pollen to generate F₁s, and then F₁s were successively backcrossed for eight generations with Zheng58$^{ga2}$ and Chang7-2$^{ga2}$ as the recurrent parental lines, respectively. A single-seed descent approach was used during backcrosses as described[8]. Four polymorphic markers M3, M5, M8 and M9 located in the $Ga2$ interval were used to select the $Ga2$ locus. After eight rounds of backcrosses, the heterozygotic individuals containing the $Ga2$ locus were selfed to obtain the homozygotic lines Zheng58$^{Ga2-S}$ and Chang7-2$^{Ga2-S}$.

**Immunolabeling analysis**. Zheng58$^{Ga2-S}$ and Zheng58$^{ga2}$ were used to observe the degree of methyl esterification (DM) of pollen tube walls. Zheng58$^{Ga2-S}$ and Zheng58$^{ga2}$ were pollinated with themselves or with reciprocal pollen, the four HAP (hour after pollination) silks were collected. The in vivo pollen-tube growth assay was performed based on the report[8]. Briefly, the silks were collected and fixed in 4% paraformaldehyde in 50 mM sodium phosphate, pH 7.0 (PBS) for at least one day. The fixed silks were washed with the above buffer and immersed in 1 M NaOH overnight for softening. The silk was then stained with 0.1% aniline blue (Amresco) in 0.1 M potassium phosphate (pH 8.0) for 6 h. The samples were observed under an Axio Skop2 microscope (Zeiss) equipped with an ultraviolet filter. Position of apical region of pollen tube in the silk was recorded, and the silk-section containing the apical region of pollen tube was collected and paraffin embedded. Thick slices of 10 μm were cut longitudinally along the silks using a microscope with a microtome (RM2235, Leica) for immunolabeling. After dewaxed by xylene solution, the silk sections were treated by an ethanol gradient to rewater and rinsed in paraformaldehyde. The sample then probed with LM20, an antibody against highly esterified homogalacturonan (University of Leeds, UK), at a 1:20 dilution[21] and FITC-coupled anti-rat IgG (Sigma, F1763) was used as a secondary antibody at a 1:500 dilution. At last, fluorescence signals were recorded with a confocal laser scanning microscope (Axio imager Z2, Zeiss). And confocal settings were the same for acquisition of every image. FITC was visualised using a 488 nm laser and 510–550 nm emission filters. The pollen tube was then defined as "tip" and "apical region"[58]. Specifically, we defined about 5 μm above the pollen tube tip point as apical region. The immunodetected fluorescence signal intensity of apical region (ROI, region of interest) was processed and analyzed using the ImageJ software (http://rsb.info.nih.gov/ij/). Silks from at least three individual plants were employed for analysis.

**Yeast two-hybrid assays**. Yeast two-hybrid experiments were performed using Matchmaker system according to the manufacturer's instructions (Clontech, https://www.clontech.com/). The coding sequences of ZmGa2P and ZmGa2F were fused in-frame with the GAL4 transcription-activation domain (AD) in the yeast vector pGADT7 (Clontech). The coding sequences of ZmGa2F, ZmPME10-1 and different domains of ZmPME10-1 were fused in-frame with the GAL4 DNA-binding domain (BD) in the yeast vector pGBKT7 (Clontech). The bait plasmid and the prey plasmid were co-transformed into the yeast strain AH109 (Clontech). Yeast transformants were screened on Synthetic Dextrose (SD) Minimal Medium with glucose but lacking leucine, tryptophan, and histidine (SD/-Leu/-Trp/-His) in dark at 28 °C for 2–3 days. The PCR-amplified primers of full-length ZmGa2P, ZmGa2F, ZmPME10-1 and different domains of ZmPME10-1 were listed in Supplementary Table 4.

**Field tests of the Ga2 locus in reproductive isolation**. The near isogenic line Chang7-2$^{Ga2-S}$ was crossed to Zheng58$^{Ga2-S}$ for generating hybrid Zheng58$^{Ga2-S}$/

Chang7-2$^{Ga2\text{-}S}$. Zheng58$^{Ga2\text{-}S}$/Chang7-2$^{Ga2\text{-}S}$ and Zheng58$^{ga2}$/Chang7-2$^{ga2}$ were subjected to compatibility test in farms located in Beijing and Sanya, Hainan province. The near isogenic hybrids were pollinated with pollen mixture of a ga2 line ZYM1 (purple kernel) and the hybrid of Zheng58$^{Ga2\text{-}S}$/Chang7-2$^{Ga2\text{-}S}$. The outcrossing rate was evaluated by the kernel color on the Zheng58$^{Ga2\text{-}S}$/Chang7-2$^{Ga2\text{-}S}$ ears compared with that of Zheng58$^{ga2}$/Chang7-2$^{ga2}$.

**Pyramiding of Ga1 locus and Ga2 locus.** Since the female determinant of the Ga2 locus was of sporophytic nature, we generated $F_1$ (Zheng58/511L) and pollinated with SDGa25 (Ga1/Ga1). We used four polymorphic DNA markers M3, M5, M8 and M9 to screen the heterozygotic plants at Ga2 locus and then selfed the ones. Markers M3, M5, M8. and M9 from Ga2 locus and marker ZmGa1P-Del from Ga1 locus[8] were applied to identify double homozygous plants at Ga1 and Ga2 loci. Eventually, we bred four lines containing double homozygous alleles at Ga1 and Ga2 loci through four rounds of selfing at two locations (Beijing and Sanya, Hainan province). For each generation, Ga1Ga2 double homozygous plants were selfed (Ga1/Ga2) and pollinated with pollen of SDGa25 (Ga1), 511L (Ga2), pollen mixture of SDGa25 and 511L (Ga1 + Ga2), and Zheng58 (ga2), respectively. All lines can only accept self-pollen while excluding pollen from either Ga1 or Ga2.

**Statistics and reproducibility.** The experiments of PAGE (polyacrylamide gel electrophoresis) were performed at least twice and similar results were obtained, thus the representative results were shown in Fig. 2d, Supplementary Figs. 4a, b and 2a, b. The genotyping of the cross-pollinated seeds in Fig. 2d were tested by at least 30 individuals. The genetic analysis experiments in Supplementary Figs. 4a, b and 2a, b were performed from 88 individuals. Statistical analysis was performed using an unpaired two-tailed student t-test and one-way ANOVA Tukey's multiple comparisons test (indicated in each figure legend). Statistical analysis and graphics were carried out with GraphPad Prism software, SPSS and Microsoft Excel.

**Reporting summary.** Further information on research design is available in the Nature Research Reporting Summary linked to this article.

## Data availability
Data supporting the findings of this work are available within the manuscript and its Supplementary Information files. The RNA-seq datasets are available from the National Center for Biotechnology Information, the BioProject and SRA accession numbers are PRJNA808835 and SRR18086820-SRR18086823. Source data are provided with this paper.

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

## Acknowledgements

We thank Prof. Weicai Yang and Dr. Hongju Li for critical discussion. This study was supported by grants from the National Natural Science Foundation of China (32172058 and 32101725).

## Author contributions

H.C. and J.L. designed and supervised the project. H.C., J.L., Z.C., and Z.Z. wrote the paper with the inputs from all authors. H.Z. performed RNA-seq analysis and *de no* transcript alignment. Z.C. and Z.Z. performed genetic studies, mapping and cloning, biochemical and immune labeling assays, gene expression, transformation analyses and isogenic line development. H.C. and J.L. analyzed the data and interpreted the results. K.L., D.C., and L.Z. performed mapping and cloning.

## Competing interests

The authors declare no competing interests.
