## [Peer Review File · Nature Communications]

A pair of non-Mendelian genes at the Ga2 locus confer unilateral cross-incompatibility in maizeReviewers' Comments:

Reviewer #1:

Remarks to the Author:

Key Results: The key results from this paper are the identification and validation of two genes for the Ga2S phenotype. Both the female and male determinants are pectin methylesterases, similar but unique from the other UCI loci. Further, the authors have assays such as yeast-2-hybrid and immunolabeling which begin to address how these PME function in gametophytic incompatibility. Lastly, they have developed a resource showing the utility of stacking Ga1 and Ga2 together in creating a pollen barrier that will evade Ga2-M lines, which they also show to be prevalent in released germplasm.

Validity: The methods and interpretations of the data are sound.

Significance: The identity of the determinants for maize gametophytic incompatibility genes has been of great interest for nearly 100 years. The identification of the Ga1 female determinant as a PME in 2017 has led to a rush of articles identifying the other UCI genes. This article fills in the last of the UCI systems. It both supports the work on the other systems while opening new avenues of research into understanding how these systems function and how they can be used effectively in commercial breeding programs.

Data and Methodology: The methods are standard, well done and the data is consistent with published or expected results. I have experience with all their methods and find them to be appropriately used and well carried out.

Suggested Improvements: I have no suggestions for additional experiments or data. I feel they have presented more than enough data to address their topic. Suggestions for minor corrections are at the end of my review.

Clarity and Context: All sections of the manuscript are clearly presented. The introduction provides adequate context for this area of research, the results are presented clearly and the methods could be followed by another researcher. The references adequately cover previous work and do a good job of addressing the current understanding in the community and the existing gaps. I have minor comments on grammar and word choices. While I would like to see those edits made, making no changes would not affect a reader's understanding of the work.

Your expertise: There was no area of the work that was outside the scope of my expertise.

My summary: I found this to be an excellent paper addressing an important topic. I am familiar with other work by these authors. I have been impressed with the quality and thoroughness of their work in the past, and this article is no different. I am excited to see this article published and discuss its findings with my colleagues.

General comment: Since this paper has the opportunity to name these genes, I wonder if the authors would consider calling the female determinant something other than ZmGa2S? The male determinant as ZmGa2P makes sense and is distinct from the phenotype (Ga2-M). It also is consistent with ZmGa1P. Perhaps the female determinants could have an -F designation, similar to Tcb-f? This becomes relevant when talking to breeders who are not well versed in genetics and making distinctions between a phenotype or haplotype (Ga2-Strong) and a gene.

Minor revisions:

Title: "the Ga2 locus"

P1 L21: "A significantly higher degree"

P8 L18: cite reference for Mo17 being Ga2-M

P8 L23: both purple and yellow kernels (currently implies a color mixture within one kernel)

P9 L13: first time using the term incompatible, make clear that the pollen is ga2 and silks are Ga2

P11 L20: determinant and be self-incompatible

P14 L8: are under way

P14-15 L27-1: correct grammar "If the male determinant was gametophytic, only one genotype (Ga2/Ga2) would be present in the 511L (♀)×(W22×511L) (♂) BC1F1 population, otherwise if it were sporophytic then two genotypes (Ga2/Ga2 and Ga2/ga2) would be present."

P15 L20: correct grammar "If the female determinant was gametophytic in nature, the BC1F1

population should have only one genotype $ga2/ga2$, otherwise if were sporophytic, there would be two genotypes $Ga2/ga2$ and $ga2/ga2$."

P18 L18: self-incompatible not self-incompatibility

P18 L23-24: change to "while heterozygous plants set both purple and yellow kernels"

P18 L25: change "mix-colored" to "a mix of colored"

P19 L3-4: grammar corrections

"and the other is a single pollination with a mixture of both pollen genotypes. Besides the full yellow and the mixed color ears, barren ears were also observed due to some individuals which lack the male determinant"

Figure 1b. Scaling is off on top B73 (152.2-153.5) chromosome. Perhaps happened in formatting?

Supplementary Figure 3: title should be "expressed in $Ga2$ pollen" (3c has expression in other $Ga2$ lines not just 511L)

Supplementary Figure 5 and 10: titles should be "predicted protein sequence"

Supplementary Figure 6: title should be "expressed in $Ga2$ -S silks" (same comment as Sup Fig 3)

Supplemental Table 1: Title needs to make clear what tissue the DEG are from (silk?)

Reviewer #2:

Remarks to the Author:

This is an excellent manuscript, and contributes valuable knowledge concerning the Unilateral Cross-Incompatibility systems in maize. They are important because of the value to the popcorn industry, as well as organic/non-gmo markets.

There are several significant contributions:

- a) confirming that there are separable male and female determinants in the $Ga2$ locus & finding recombinants separating male and female functional determinates,
- b) determining potential functional genes by mapping & following with expression data matching the phenotypic observations to identify the gene,
- c) that the female function is conditioned by a single recessive gene and that it is of sporophytic operation,
- d) map-based cloning and identification of the likely female gene, and
- e) proving that no direct interaction between the silk function determinate and the male function determinate is involved in their phenotypic operation.

The materials, methods, and the data are of good quality and clearly support the conclusions reached.

I am enclosing some language/form suggestions that reflect standard English usage, however, the paper is very well written, and is much better than I could do in a language which isn't my first.

Page 2, line 4 ...has been "known" for more...

Page 2, line 10 ...a pair of female... (leave out second "a")

Page 2, line 28 ... of UCI, "exploration" , not exploitation

Page 2, line 29 ...major "progress has"...

page 3, line 4 (divide the sentence) ...in a $ga1$ background. These plants can...

page 3, line 19 & 20 ..."progress in this area has been slower" than that in...

page 5, line 4 ...that only pollen of the... POLLEN IS SINGULAR

page 5, line 15 ..or sporophytic, a (W22x....

page 5, line 16 ... population was developed... , not were

page 7, line 25 I would divide the sentence. ...four $ga2$ lines). The two $Ga2$

page 10, line 3 ...the interaction, but no.. (add , use but)

page 11, line 20 ...determinant and be self-incompatible....

I am pleased with the methodology, the data presented, and the conclusions reached in your paper. It adds to what is known about all three of the UCI loci in maize.

Reviewer #3:

Remarks to the Author:

The manuscript of Chen and coworkers reports a very interesting study of the Ga2 locus, one of the loci involved in the unilateral cross incompatibility (UCI) system of maize and one of the lesser known. The authors perform a robust genetic and molecular study of the male and female determinants of Ga2, both of them encode pectin methyl esterases (PME), responsible of de-esterification of pectins. It is known that PME activity regulates the level of pectin esterification in cell walls and that the dynamic cell wall of the tip of growing pollen tube is enriched in highly methyl esterified pectins, while other parts of the tube wall contain major proportion of de-esterified pectins. On the other hand, in maize it has been postulated that PME activity in the growing pollen tube tip has a critical role in controlling maize UCI, by regulating the degree of methylesterification of the pollen tube cell wall, i.e. low esterification involves no progression of the tube and no fertilization.

Functional analyses of the manuscript are based in the study of the level of pectin esterification in pollen tube tips from different genetic crosses, growing in compatible and incompatible silks. Results indicate that pollen tube tips contain less esterified pectins in incompatible silks than in compatible ones. These findings together with other results of the manuscript lead authors to conclude that ZmGa2P determinant may not possess *in vivo* PME activity and to hypothesize that it may interact with other active PMEs that have both PME and PMEI domains to finely tune PME activities in the apical region of pollen tube.

The main support for the conclusion of the manuscript regarding the function of the Ga2 determinants is the different esterification level of pectins of the pollen tube tips in compatible versus incompatible silks. This analysis is based in a unique experimental approach: immunofluorescence with LM20 antibody which detects highly esterified pectins, and quantification of signal intensities. Since no other approach is performed to estimate changes in pectin esterification, the methodology and results of these assays should be clearly explained. Some concerns in this sense:

Immunofluorescence experiments with LM20 in pollen tubes growing *in vitro* would be interesting to compare with *in vivo* situation. This comparison would help to understand the function of female determinant and to give additional support to the results *in vivo*.

Page 9, line 10: Title is confuse, the genes are PMEs, and pectins are synthesized as highly esterified forms that can be de-esterified later.

Page 9, lines 22-24: a clear definition of the region considered as "tip", "apical region" of the pollen tube should be included (length, shape). Images at lower magnification, showing pollen tube regions with no LM20 signal would support the results.

For an appropriate comparison among immunofluorescence signal intensities methodology should be clear. In M&M or Results section, a short description on the method used for quantification of signals is required (software used, number of images, type of images (optical sections, projections, z-stack collection...) also, definition of ROI (region of interest) to quantify the signal, and if confocal settings were the same for acquisition of every image, etc.

What do dotted lines mean in figure 5?

Ideally, immunofluorescence assays with other antibody which detects pectins with low degree of esterification (e.g. LM19) could be added to the manuscript, to support the results with LM20 (or any other experiment that can provide additional support).

Reviewer #1:

Q1: Since this paper has the opportunity to name these genes, I wonder if the authors would consider calling the female determinant something other than ZmGa2S? The male determinant as ZmGa2P makes sense and is distinct from the phenotype (Ga2-M). It also is consistent with ZmGa1P. Perhaps the female determinants could have an -F designation, similar to Tcb-f? This becomes relevant when talking to breeders who are not well versed in genetics and making distinctions between a phenotype or haplotype (Ga2-Strong) and a gene.

RE: Great suggestion! We have renamed the female determinant as ZmGa2F throughout the text.

Q2: Title: “the Ga2 locus”

P1 L21: “A significantly higher degree”

RE: Revised (P1 L1, L21). Thanks!

Q3: P8 L18: cite reference for Mo17 being Ga2-M

P8 L23: both purple and yellow kernels (currently implies a color mixture within one kernel)

RE: The reference was added (P8 L17) and the sentence was revised (P8 L22).

Q4: P9 L13: first time using the term incompatible, make clear that the pollen is ga2 and silks are Ga2

RE: Detailed description was added as suggested (P9 L13).

Q5: P11 L20: determinant and be self-incompatible

RE: Revised (P11 L19).

Q6: P14 L8: are under way

P14-15 L27-1: correct grammar “If the male determinant was gametophytic, only one genotype (Ga2/Ga2) would be present in the 511L (♀)×(W22×511L) (♂) BC1F1

population, otherwise if it were sporophytic then two genotypes (Ga2/Ga2 and Ga2/ga2) would be present.”

RE: Corrected (P14 L5, L25-27).

Q7: P15 L20: correct grammar “If the female determinant was gametophytic in nature, the BC1F1 population should have only one genotype ga2/ga2, otherwise if were sporophytic, there would be two genotypes Ga2/ga2 and ga2/ga2.”

RE: Corrected (P15 L17-18).

Q8: P18 L18: self-incompatible not self-incompatibility

P18 L23-24: change to “while heterozygous plants set both purple and yellow kernels”

P18 L25: change “mix-colored” to “a mix of colored”

RE: Appreciated! All were revised (P18 L16, L22, L23).

Q9: P19 L3-4: grammar corrections

“and the other is a single pollination with a mixture of both pollen genotypes. Besides the full yellow and the mixed color ears, barren ears were also observed due to some individuals which lack the male determinant”

RE: Corrected (P19 L1-3).

Q10: Figure 1b. Scaling is off on top B73 (152.2-153.5) chromosome. Perhaps happened in formatting?

Supplementary Figure 3: title should be “expressed in Ga2 pollen” (3c has expression in other Ga2 lines not just 511L)

Supplementary Figure 5 and 10: titles should be “predicted protein sequence”

Supplementary Figure 6: title should be “expressed in Ga2-S silks” (same comment as Sup Fig 3)

Supplemental Table 1: Title needs to make clear what tissue the DEG are from (silk?)

RE: All were revised as suggested.

Reviewer #2:

Q1: Page 2, line 4 ...has been "known" for more...

Page 2, line 10 ...a pair of female... (leave out second "a")

Page 2, line 28 ... of UCI, "exploration", not exploitation

Page 2, line 29 ...major "progress has"...

RE: Thanks. All were corrected (Page 2, lines 4, 10, 28, 29).

Q2: page 3, line 4 (divide the sentence) ...in a gal background. These plants can...

page 3, line 19 & 20 ..."progress in this area has been slower" than that in...

RE: Thanks. Revised as suggested (Page 3, lines 5, 19).

Q3: page 5, line 4 ...that only pollen of the... POLLEN IS SINGULAR

page 5, line 15 ...or sporophytic, a (W22x....

page 5, line 16 ... population was developed... , not were

RE: All of the above were corrected (Page 5, lines 2, 14, 15).

Q4: page 7, line 25 I would divide the sentence. ...four ga2 lines). The two Ga2....

RE: Thanks. We have divided the sentence as suggested (Page 7, line 24).

Q5: page 10, line 3 ...the interaction, but no.. (add , use but)

RE: Revised as suggested (Page 10, line 2).

Q6: page 11, line 20 ...determinant and be self-incompatible....

RE: Corrected (Page 11, line 19).

Reviewer #3:

Q1: Immunofluorescence experiments with LM20 in pollen tubes growing *in vitro* would be interesting to compare with *in vivo* situation. This comparison would help to

understand the function of female determinant and to give additional support to the results *in vivo*.

RE: This is a good point and we totally agree that an *in vitro* assay would give additional support. However, *in vivo* assay would reflect the actual in-plant situation. In fact, we have tried to express ZmGa2P and ZmGa2F *in vitro* using both *Escherichia coli* and *Pichia methanolica* expression systems. Unfortunately, we failed to detect any PME activity in the assay reaction, thus no active proteins were obtained for the *in vitro* immunofluorescence experiments. Similar situation happened for ZmGa1P. We proposed the possible reason as the lack of post-translational modification in these expression systems. This provides an important topic for future studies.

Q2: Page 9, lines 10: Title is confused, the genes are PMEs, and pectins are synthesized as highly esterified forms that can be de-esterified later.

RE: We revised the title as “ZmGa2P and ZmGa2F are putative PMEs mediating cross-incompatibility” (Page 9, Lines 9).

Q3: Page 9, lines 22-24: a clear definition of the region considered as “tip”, “apical region” of the pollen tube should be included (length, shape). Images at lower magnification, showing pollen tube regions with no LM20 signal would support the results.

RE: The “tip” and “apical region” both indicate the region shown as the figure below. Detailed description was included in the M&M section (Page 20, Lines 4-7). Maize silks are usually up to ~10 cm long, images showing pollen tube regions are very difficult to take even at lower magnification. Sectioned silks are often used for pollen tube growth study in maize.

Guan, Y., Guo, J., Li, H. & Yang, Z. Signaling in pollen tube growth: crosstalk, feedback, and missing links. *Mol. Plant* **6**, 1053-1064 (2013).

Q4: For an appropriate comparison among immunofluorescence signal intensities methodology should be clear. In M&M or Results section, a short description on the method used for quantification of signals is required (software used, number of images, type of images (optical sections, projections, z-stack collection...) also, definition of ROI (region of interest) to quantify the signal, and if confocal settings were the same for acquisition of every image, etc.

RE: Thanks. We have updated and revised both M&M section (Page 19, lines 17-25; Page 20, lines 1-10) and the Figure 5 legend as suggested.

Q5: What do dotted lines mean in figure 5?

RE: The dotted lines are the arbitrary division lines between the apical region and the rest of the pollen tube. Explanation is added in the figure legend.

Q6: Ideally, immunofluorescence assays with other antibody which detects pectins with low degree of esterification (e.g. LM19) could be added to the manuscript, to support the results with LM20 (or any other experiment that can provide additional support).

RE: We fully agree that immunofluorescence assays with additional antibody could provide additional support for LM20 results. This study mainly focused on the genetic

study, map-based cloning and functional validation of the *Ga2* locus, which laid foundation for subsequent molecular, genetic and biochemical studies. Additional immunofluorescence assays with other antibody are necessary for future study.

Reviewers' Comments:

Reviewer #1:

Remarks to the Author:

I am satisfied with the revisions made by the authors. Upon another reading, I did find a few more grammatical or spelling errors. I also have a comment about the introduction (P3 L9-17).

P1 L25: as a reproductive

P1 L26: programs

P2 L10: by a pair of female

P3 L5: functional studies

P3 L13: ZmPme3 paper used de novo RNAseq, and was published two years before the Tcb-f paper and one year before the Ga1P paper. It is the first report of PME involvement in maize UCI, and its publication inspired the use of de novo RNAseq in the Tcb work, and should be credited as such.

P3 L15: functionally validated

P3 L23: eventual deciphering of the

P5 L16-17: either "plants would produce full seed-set ears" or "plants would set full ears"

P10 L4: pollen tubes incompatible crosses

P13 L5-6: may be secreted extracellularly

P17 L22: cut the first 15-bp

Reviewer #2:

Remarks to the Author:

There are a few clarity/grammar/usage suggestions I found on reading/reviewing the paper a final time:

page 5, line 26 The same... rather than Same...

page 6, lines 19&20 upstream from... and downstream from..., rather than upstream...,downstream....

page 12, line 24 This type of protein ... (singular, rather than proteins, plural)

Reviewer #3:

Remarks to the Author:

Authors have addressed most of my concerns and the manuscript has improved.

Reviewer #1:

Q1: P1 L25: as a reproductive

P1 L26: programs

P2 L10: by a pair of female

P3 L5: functional studies

RE: All are corrected. Thanks!

P3 L13: ZmPme3 paper used de novo RNAseq, and was published two years before the Tcb-f paper and one year before the Ga1P paper. It is the first report of PME involvement in maize UCI, and its publication inspired the use of de novo RNAseq in the Tcb work, and should be credited as such.

RE: Thanks. Revised as suggested.

P3 L15: functionally validated

P3 L23: eventual deciphering of the

P5 L16-17: either “plants would produce full seed-set ears” or “plants would set full ears”

P10 L4: pollen tubes incompatible crosses

P13 L5-6: may be secreted extracellularly

P17 L22: cut the first 15-bp

RE: All are revised. Thanks.

Reviewer #2:

page 5, line 26 The same... rather than Same...

page 6, lines 19&20 upstream from... and downstream from..., rather than upstream.....downstream....

page 12, line 24 This type of protein ... (singular, rather than proteins, plural)

RE: Thanks. All of the above were corrected.

Reviewer #3:

Authors have addressed most of my concerns and the manuscript has improved.

RE: Thanks again for your input.